

# Drier spring over the US Southwest as an important precursor of summer droughts over the US Great Plains

Amir Erfanian[1*] and Rong Fu[1]

[1] Department of Atmospheric and Oceanic Sciences, University of California Los Angeles, CA,
USA

* Correspondence:
A. Erfanian
Department of Atmospheric and Oceanic Sciences
University of California Los Angeles, Los Angeles, CA, 90095
Email: amir.erfanian@atmos.ucla.edu
Phone: 860 402 7756

**Abstract**

This study addresses the role of atmospheric moisture budget in determining the onset and
development of the summer droughts over the North American Great Plains (GP) using two
state-of-the-art reanalysis datasets. We identified zonal moisture advection as the major cause of
the severe tropospheric drying during the extreme droughts of southern GP 2011 and northern
GP 2012. For both events, an eastward advection of anomalously dry and warm air in the free
troposphere in spring sets the stage for the summer drought, leading to a sharp drop of relative
humidity above the boundary layer, enhancing dry entrainment, and suppressing deep
convection. Further breakdown of the zonal advection into the dynamic (caused by circulation
anomalies) versus thermodynamic (caused by moisture anomalies) contributions revealed
dominance of the thermodynamic advection in the tropospheric drying observed during the onset
of both 2011 and 2012 droughts. The dependence of thermodynamic advection on moisture
gradient links the spring precipitation in the Rockies and southwestern US, the source region of
the anomalous dry advection, to the GP summer precipitation (with correlations > 0.4 using
gauged-based data). The results of this study improve our predictive understanding of summer
drought onset mechanisms over the GP.

Key words: drought, precipitation, moisture budget, thermodynamic and dynamic advection,
Great Plains, US southwest




1. **Introduction**

The United States (US) Great Plains (GP) is prone to devastating droughts, including the infamous "Dust Bowl" of the 1930s (e.g. Brönnimann et al. 2009; Donat et al. 2016), the extended drought in the 1950s (e.g. Cook et al., 2011), the Texas drought of 2011 (e.g. Fernando et al. 2016), and the record-breaking drought of 2012 (e.g. Hoerling et al., 2013). The projections of the Global Climate Models (GCMs) that participated in the Coupled Model Intercomparison Project Phase 5 (CMIP5) show a robust intensification of dry conditions over the GP under different global warming scenarios in the coming decades (Cook et al., 2015; Teng et al., 2016), which would damage the agricultural and food industries over the region. The current dynamic prediction models have virtually zero prediction skill over the GP in summer (Quan et al., 2012; Hoerling et al., 2014). Improvement of our predictive understanding of drought onset and evolution mechanisms would provide scientific foundation for a more accurate and timely prediction of droughts over the region.

The GP drought and its underlying mechanisms have been studied extensively. Numerous studies have shown that, in the early stages of the GP droughts, the upper-level atmosphere has featured an anomalous high and anticyclonic vorticity over central North America (Chang and Wallace, 1987; Namias, 1991; Lyon and Dole, 1995; Cook et al., 2011; Donat et al., 2016; Fernando et al., 2016). A dynamical teleconnection between the height anomalies over the US and the North Pacific SST anomalies has been considered as the main driver responsible for the onset of GP summer droughts in 1980 and 1988 (Trenberth, et al., 1988; Lyon and Dole, 1995; Chen and Newman, 1998). Variability of Pacific and Atlantic SSTs has been considered an important driver of droughts in North America with warm SST anomalies in tropical Atlantic and cold SST anomalies in tropical and eastern North Pacific favoring summer droughts over the GP (Namias, 1991; McCabe et al., 2004; Schubert et al., 2004; Kushnir et al., 2010; Wang et al., 2010; Feng et al., 2011; Zhao et al., 2017). However, the role of SST as a main driver of GP precipitation variability has been challenged by numerous studies arguing that the atmospheric internal variability and land-atmosphere feedbacks are the dominant drivers of the GP summer drought for both short (Hoerling et al., 2013; Wang et al., 2014; Fernando et al., 2016; Pu et al., 2016) and long-term (Schubert et al., 2004; Ferguson et al., 2010) time scales. Despite the extensive research, it has remained unclear whether the SST anomalies in winter and spring can significantly influence summer GP droughts. If so, what are the underlying physical mechanisms? This question is central in determining whether the GP summer drought is predictable. As a first step, we need to understand the main cause of the moisture deficit that initiate the summer droughts in the GP. To our knowledge, a systematic moisture budget analysis to determine such causes previously has not been reported.

Moisture budget analysis has been attempted previously to understand local and large-scale sources of moisture (Rasmusson, 1968; Yanai et al., 1973). The climatology, seasonal and diurnal cycles of moisture budget terms have been analyzed in the work of (Rasmusson, 1968) over the US and in studies of (Hao, 1987; Zangvil et al., 1993, 2001; Schubert et al., 1998; Lamb et al., 2012) over the US GP. For the Southern GP (SGP), Lamb et al. (2012) calculated the



vertically integrated Moisture Flux Convergence (MFC) terms using the North American
Regional Reanalysis in May-June for four selected years (1998, 2002, 2006, and 2007) and
identified the horizontal advection and divergence terms, respectively, responsible for the
moisture transport to and from the SGP. For the 1980 drought, Hao (1987) compared the
vertically integrated horizontal advection and divergence terms of 1980 and 1979 summers
(calculated from radiosonde data over the SGP) and indicated that the horizontal divergence was
the dominant contributor to the extreme drying in the summer of 1980. Schubert et al., (1998)
identified the GP low-level jets (LLJ) as the dominant contributor to the summer mean moisture
influx to the US interior and indicated a strong link between sub-seasonal variability of moisture
influx from the Gulf of Mexico and warm-season precipitation over the central and eastern US.
While these studies provide very useful information about the atmospheric moisture sinks and
sources over the GP, they only focused on the warm-season vertically integrated budget terms in
a few selected years/periods in their analysis. Investigating the vertical structure of the individual
moisture tendencies, their seasonal evolution and year-to-year variability, and the relative
importance of the moisture transport and evapotranspiration (ET) anomalies on precipitation
variability are important questions that need to be addressed yet to understand the processes and
feedbacks underlying the GP droughts specially during the onset season (March, April, May).

    In this paper, we provide a detailed examination of the atmospheric moisture budget terms
using two state-of-the-art reanalysis datasets over the entire period of 1980-2018 (see section 2
for details). Our diagnostic analyses present a comprehensive picture of GP tropospheric
moisture sinks/sources by investigating the diurnal cycle and the vertical structure of moisture
budget terms and their temporal evolution before and during extreme droughts. A unique
contribution of our study is the determination of physical processes that control the variability of
moisture tendency over the GP. This was achieved by separating the moisture transport
anomalies into their thermodynamic and dynamic contributions, identifying the regional and
remote drivers that modulate variability of these contributions, and measuring the relative
importance of the individual terms in the onset and development of GP droughts. In the rest of
the paper, we provide a detailed explanation of the implemented methods in section 2, present
the results and discussion in section 3, and provide a summary of the results and our main
conclusions in section 4.

## 2. Methodology

### 2.1. Moisture budget

Combining the mass continuity equation ($\nabla.\boldsymbol{v} + \frac{\partial \omega}{\partial p} = 0$) with conservation of water vapor
($\frac{Dq}{Dt} = S$), Eq. 1 presents the water vapor budget equation for a unit mass of air (Yanai et al.,
105  1973):

$$\frac{\partial q}{\partial t} + \nabla.(q\boldsymbol{v}) + \frac{\partial(q\omega)}{\partial p} = e - c \qquad (1)$$



where t, q, and p stand for time, specific humidity, and pressure respectively; $\mathbf{v}$ and $\omega$ are the horizontal wind vector and vertical wind velocity in pressure coordinate, and e and c are the evaporation and condensation rates of the air parcel per unit mass, respectively.

Assuming a negligible contribution from the moisture tendency term (the 1st term in Eq. 1) for monthly and longer time averages (Trenberth and Guillemot, 1995), vertical integration of Eq. 1 from Pt=0 to Ps results in:

$$-\int_0^{P_s} \mathbf{v}.\nabla q\ dp - \int_0^{P_s} \omega \frac{\partial q}{\partial p}\ dp = g\rho_w(P - E) \qquad (2)$$

where P and E are precipitation and evapotranspiration rates at the surface and g and $\rho_w$ stand
for gravitation acceleration of the Earth and water density. Decomposing an arbitrary variable A to a stationary ($\tilde{A}$) and a transient term ($\acute{A}$) ($A = \tilde{A} + \acute{A}$), and applying the covariance equation ($\widetilde{qv} = \tilde{q}\tilde{v} + \widetilde{\acute{q}\acute{v}}$), we can write Eq. 2 as the following:

$$-\int_0^{P_s} \tilde{u}\ \partial_x\tilde{q}\ dp - \int_0^{P_s} \tilde{v}\ \partial_y\tilde{q}\ dp - \int_0^{P_s} \tilde{\omega}\frac{\partial\tilde{q}}{\partial p}\ dp\ - \int_0^{P_s} (\partial_x\widetilde{\acute{q}\acute{u}} + \partial_y\widetilde{\acute{q}\acute{v}} + \frac{\partial\widetilde{\acute{q}\acute{\omega}}}{\partial p})\ dp$$
$$= g\rho_w(P - E) \qquad (3)$$

where u and v are the zonal and meridional components of the horizontal wind, v. The first, second, and third terms on the left hand side (LHS) represent the zonal, meridional, and vertical advection, respectively and the last term in the LHS refers to the eddy transient terms of the zonal, meridional, and vertical winds. In our analysis, the transient and stationary terms refer to the monthly mean and six-hourly departure from the monthly mean (see sections 2.3 and 2.4 for
more information on the temporal and spatial resolution of the input data and numerical calculations).

## 2.2. Thermodynamic versus dynamic contribution

Breaking up each term in Eq. 3 to a climatological mean and a monthly departure from climatology (e.g. $\tilde{A} = \bar{A} + \acute{A}$), Eq. 3 can be organized as the following (Chou and Lan, 2012; Li
et al., 2016; Peng and Zhou, 2017):

$$P' = -\frac{1}{g\rho_w}\left(\int_0^{P_s} u\ \partial_x q\ dp\right)' - \frac{1}{g\rho_w}\left(\int_0^{P_s} v\ \partial_y q\ dp\right)' - \frac{1}{g\rho_w}\left(\int_0^{P_s} \omega\frac{\partial q}{\partial p}\ dp\right)' + E'$$
$$+ \varepsilon' \qquad (4)$$

where, the anomalous precipitation is balanced by the anomalous advection, evaporation, and residual $\varepsilon$ which accounts for the sub-monthly transient eddy contribution. The transient
advection terms in Eq. 4 can be further separated as

$$-\left(\int_0^{P_s} u\ \partial_x q\ dp\right)' \approx -\int_0^{P_s} \bar{u}\ \partial_x q'\ dp - \int_0^{P_s} u'\ \partial_x \bar{q}\ dp - \int_0^{P_s} u'\ \partial_x q'\ dp \qquad (5)$$



The first term in the right hand side (RHS) of Eq. 5 is referred to as the thermodynamic contribution of the zonal advection which accounts for the changes in humidity while setting the circulation to climatological wind. The second term in the RHS is referred to as the dynamic

contribution which accounts for the changes in wind given the climatological humidity and the third term in RHS is the non-linear term which accounts for the interannual anomalies of both wind and humidity (Seager et al., 2010; Chou and Lan, 2012; Li et al., 2016; Peng and Zhou, 2017). Separating all the advection terms into thermodynamic and dynamic contributions, Eq. 4 can be rewritten as the following:

$$P' = -\frac{1}{g\rho_w}\int_0^{P_s}(\bar{u}\,\partial_x q' + u'\,\partial_x\bar{q} + u'\,\partial_x q')dp - \frac{1}{g\rho_w}\int_0^{P_s}(\bar{v}\,\partial_y q' + v'\,\partial_y\bar{q} + v'\,\partial_y q')dp$$
$$-\frac{1}{g\rho_w}\int_0^{P_s}\left(\bar{\omega}\frac{\partial q'}{\partial p} + \omega'\frac{\partial\bar{q}}{\partial p} + \omega'\frac{\partial q'}{\partial p}\right)dp + E' + \varepsilon' \quad (6)$$

### 2.3. Data

The moisture budget analysis in this study is based on the European Centre for Medium-Range Weather Forecasts (ECMWF) Interim Re-Analysis (ERA-Interim) (Dee et al., 2011) which

covers 6-hourly upper air parameters from 1979 to near-real-time. The atmospheric model has 60 levels in a hybrid sigma-pressure vertical coordinate system and a T255 spectral horizontal resolution (~79 km). The data is available online (http://apps.ecmwf.int/datasets/data/interim-full-daily/levtype=sfc/). In addition to ERA-Interim, we also used the Modern-Era Retrospective Analysis for Research and Applications-version 2 (MERRA-2) (Gelaro et al., 2017) and repeated

the moisture budget analysis to ensure that our conclusions were not sensitive to the choice of the reanalysis product. MERRA-2 is the latest atmospheric reanalysis of the National Aeronautics and Space Administration (NASA) Global Modeling and Assimilation Office (GMAO) covering the 1980 to near-present time period and is available online at the NASA GMAO website (https://gmao.gsfc.nasa.gov/reanalysis/MERRA-2/). The atmospheric model uses a cubed-sphere

horizontal grid with a 0.5°x0.625° resolution and a hybrid-eta vertical coordinate system with 72 model levels from the surface to 0.01 mb.

For observed precipitation we used the National Centers for Environmental Prediction (NCEP) Climate Prediction Center (CPC) unified Gauged-based analysis of daily precipitation over the continental US with a 0.25°x0.25° resolution. The data are available from 1948 to present,

provided by National Oceanic and Atmospheric Administration (NOAA) Earth System Research Laboratory (ESRL), Physical Science Division (PSD), Boulder, Colorado, USA at https://www.esrl.noaa.gov/psd/.

### 2.4. Computation

The moisture budget terms in Eq. 3 were calculated using 6-hourly ERA-Interim reanalysis on a

regular 0.75° grid and 14 selected pressure-levels. The horizontal and vertical gradients were calculated using a centered finite difference approach. The vertical integrals were performed by integrating the product of the moisture tendencies in each layer multiplied by the pressure



thickness of each layer (dP) from surface to 50-mb level. The calculations were done at 14
pressure-levels (spanning 1000 mb to 50 mb) where the lowest 6 levels (from 1000 mb to 850
mb), which contain most of the atmospheric moisture, had a 25-mb resolution and the thickness
of the remaining levels grew to 50 mb and 100 mb for the mid and upper troposphere. The
vertically integrated moisture tendencies were divided by $g\rho_w$ and multiplied by a scale factor
$(24*3600*10^{-3})$ to convert m/s to mm/day. To determine the impacts of daytime and nocturnal
anomalous circulation, we have separately computed daytime and nighttime composites. The
daytime composites for the North American domain were calculated by averaging the reanalysis
outputs at 1800 and 0000 Universal Time Coordinate (UTC), and the nighttime composites were
obtained by averaging the reanalysis outputs at 0600 and 1200 UTC. More information on the
benefits and limitations of the diagnostic computation of the atmospheric moisture budget with
reanalysis is provided by Kevin E. Trenberth and Guillemot (1995) and Seager and Henderson
(2013), including the impacts of several sources of errors including temporal, horizontal, and
vertical resolution, numerical calculation of gradients and vertical integration, and reanalysis
initialization.

### 2.5. Significance of correlation coefficients

There are 39 annual samples during our analysis period of 1979-2018. Accounting for the
effective sample size by using the (Livezey et al., 1983) method for a lag-1 auto-correlation of
0.2 for two time series ($r_1$=$r_2$=0.2 and $r_1r_2$=0.04) (which is a conservative estimate for the annual
time series of standardized anomalies of P, q, and zonal moisture advection) results in
significance levels of 7.1% and 1.4% for the correlation coefficients of 0.3 and 0.4, respectively
using a two-tailed Student-t distribution with N-2 degrees of freedom.

## 3. Results

### 3.1. The Great Plains Summer Drought

The US GP, located east of the Rocky mountains and west of the Mississippi River, are
characterized by a semi-arid climate with a land surface covered primarily by farmlands and
temperate grasslands. On average, the region has an annual precipitation of 1-2 mm/d,
approximately half of which occurs during the boreal summer (Figure 1b). The climatology of
observed summer precipitation during 1979-2018 features a zonally asymmetric pattern with JJA
precipitation less than 1 mm/d over the Rockies and US southwest, between 1 and 3 mm/d over
the central plains, and greater than 3 mm/d over the US Midwest and eastern US (Figure 1a). The
GP have been subject to recurrent severe drought and heat waves with two extreme droughts in
2011 and 2012 occurring during the most recent decade (Figure 1e and also Cook et al., 2011;
Hoerling et al., 2013; Fernando et al. 2016). Summer droughts over the region usually develop in
previous spring and peak in mid- to late summer. As indicated by the maps of JJA standardized
precipitation anomalies (Figures 1c, 1d), the drought of 2011 was confined to the SGP,
especially Texas, while the 2012 event covered nearly the entire US GP with the drought
epicenter over the NGP. The temporal evolution of the 2011 drought shows a steady decline of
precipitation and ET started in February, extending throughout the spring and peaking during
summer (-2 mm/d) in both MERRA2 and ERA-Interim reanalysis data (Figures 2a and 2b). The



dry anomalies started recovering in the fall and the drought finally ended in late fall/early winter. Similarly, the NGP 2012 drought developed (somewhat rapidly) in spring as noted by a sharp

decline in precipitation in March followed by a normal April and a large drop in precipitation and ET in May (Figures 2c and 2d). The negative precipitation and ET anomalies extended through the 2012 summer and early fall with a gradual recovery of drought conditions closer to winter. As shown in Figure 2, the amplitude of the anomalies and their temporal evolution is consistent between the two reanalysis datasets.

The atmospheric profiles of specific humidity ($q_v$) and cloud liquid and ice water content during the 2011 SGP and 2012 NGP droughts are compared against the 1979-2018 climatology in Figure 3 and 4 (see Figure S1 for relative humidity (RH) and Fraction of Cloud Cover (FCC)). The $q_v$ climatology for both the SGP and NGP indicates that the largest annual values occurred during summer where the maximum humidity (larger than 10 g/kg) was confined to the lower

troposphere (1000-800 mb) and gradually decreased to ~5 and ~3 g/kg in the mid- and upper-troposphere (Figures 3a and 4a). The climatological value of specific humidity at all levels start decreasing in fall with the lowest annual rates (<3 g/kg) during winter. The annual average values of the specific cloud ice and water content peak in fall and spring, and reach the minimum values during summer over both the SGP and NGP, respectively (Figures 3d and 4d). Over the

NGP, annual minimum values also occur in winter (Figure 4d). The spring-time peak of specific cloud liquid and ice water is much greater than the peak values in fall with the largest values (>10 g/kg) confined to the low- and mid-troposphere (850-500 mb) in the NGP and the lower-free troposphere (850-650 mb) in the SGP.

The specific humidity and cloud liquid and ice water in both dry years were generally much

smaller than their climatological values. However, the two variables reveal distinct temporal evolutions and vertical structures. As revealed by Figures 3c and 4c, large negative anomalies of $q_v$ extending from the surface to the mid-troposphere persisted year-around for the SGP 2011 event. The maximum dry anomalies of $q_v$ were located in the near surface levels and peaked during the May-June and August-September periods, indicating intensive drying of the boundary

layer air during the drought peak. The dry anomalies in summer was preceded by an extended drier lower troposphere in the spring season. For the 2012 NGP event, however, the spring-time anomalies of $q_v$ remained reasonably wet during March-April until the drought intensified rapidly in May. The $q_v$ anomalies remained negative during the entire summer and fall 2012 with the largest negative anomalies occurring near the surface in August and September.  Figure 4f

shows that the negative anomalies of cloud liquid and ice water content over the NGP developed in winter, and persisted during the entire 2012. The negative anomalies of cloud liquid and ice water content started four months earlier than the negative $q_v$ anomalies, highlight the impact of warmer temperature during 2012 winter and spring, which reduced relative humidity (Figure S2c) and consequently, cloud liquid and ice water (Figure 4f) and fractional coverage (Figure

S2f), as well as depleted soil moisture (Sun et al., 2015; Mo et al., 2016).

For both 2012 NGP and 2011 SGP drought years, the cloud liquid and ice water content of dry years were much lower throughout the depth of the troposphere and over the course of the





year with the largest decline (~40%) occurring in the lower- and mid-tropospheric levels in spring and early summer (Figures 3f and 4f). The drying of the low- and mid-troposphere was
linked to a sharp drop of mid- and upper-troposphere RH in spring as shown in Figures S1c and S2c. A sharp decline of free tropospheric RH intensifies the entrainment of dry air into the rising moist air above the boundary layer limiting the convective penetration depth and shifting the convection structure from predominantly deep convection to frequent shallow cumulous clouds (Derbyshire et al., 2004; Zhang et al., 2010; Del Genio, 2012). The FCC difference fields during
the spring and early summer of both 2011 over the SGP and 2012 over the NGP indicate large negative anomalies extending from above the Planetary Boundary Layer (PBL) to the upper troposphere suggesting a strong suppression of deep convection during the onset season (Figures S1f and S2f).

### 3.2. **Moisture budget analysis**

Summer in the GP is the warmest season of year with the highest rate of seasonal evapotranspiration (ET). Despite its highest share of annual rain, JJA precipitation minus evapotranspiration (P-E) is negative with the maximum deficit (1-3 mm/d) over the GP and US Midwest. Such a P-E deficit is balanced by the atmospheric moisture flux convergence (MFC) over monthly and seasonal time scales (see section 2.1). Using ERA-Interim 6-hourly data over
1979-2018, we calculated the individual moisture tendencies in Eq. 3 and compared the sum of vertically integrated terms with the ERA-Interim reported vertically averaged moisture convergence (-1*divergence) to evaluate the accuracy of our numerical calculations. The spatial patterns of the JJA climatology of the MFC are very similar between our calculated values and those reported by ERA-Interim, for example, over the inter tropical convergence zone (ITCZ)
between the equator and 15°N and over the regions of sub-tropical anticyclones (Figure 5a and 5b). Over land, the JJA climatology in the ERA-Interim MFC and that of numerically calculated MFC from the 6-hourly atmospheric fields indicate near zero differences over much of Alaska, western Canada, and central and eastern US, except for over the complex terrain of western US and north western Atlantic (Figure 5d), where a relatively larger difference (between 0.5 and 1.5
mm/d) occurs. These differences originate from multiple sources including the vertical resolution (14 pressure levels in our calculation vs the 60 model levels in ERA-Interim) and numerical calculation of the divergence and gradient terms (see Trenberth et al., 2011 and Seager and Henderson, 2013 for more details). Overall, our numerically calculated MFC maintains a desirable accuracy in comparison to the ERA-Interim MFC. The difference fields between MFC
and P-E reveal moisture budget imbalances as large as 1.5 mm/d over the US central plains in JJA for both calculations of MFC (Figures 5e and 5f). The imbalance is partially due to the (neglected) atmospheric moisture storage and in part due to the unclosed moisture budget in the reanalysis (Trenberth et al., 2011; Seager and Henderson, 2013).

To investigate the GP summer droughts from a moisture budget perspective, we looked at the
individual moisture tendencies, their vertical structure, annual cycle, and diurnal variability for both the 2011 and 2012 events compared with the 1979-2018 climatology in ERA-Interim. For the SGP (Figure 6), all climatological tendencies indicate strong seasonal variability with the



vertically integrated tendencies (blue line) revealing positive values (moisture convergence) as large as 1 mm/d for the zonal advection during summer, meridional advection year-round,

vertical advection during spring, and horizontal transient term during winter (Figures 6c, 6f, 6i, and 6l). The major sources (<-1 mm/d) of negative tendencies (moisture divergence) are the transient term in spring and summer and the vertical advection term in fall and winter. For the summer season in particular, the eddy transient term features strong moisture divergence extending from the surface to the upper troposphere (Figures 6j and 6l) whereas the meridional

advection reveals strong positive tendencies that are confined to the lower troposphere and much stronger during night featuring the moisture transport from the Gulf of Mexico through the GP LLJ (Figures 6d and 6f). The zonal advection also indicates moderate to strong positive tendencies during summer that are confined to the mid- and upper-troposphere (Figures 6a and 6c). The difference between the day-time and night-time moisture tendencies is quite large for

the meridional advection term during spring and summer and the vertical advection term during late summer and fall, and negligible year-around for the horizontal advection and transient terms. The annual cycle and the vertical structure of all moisture terms for the SGP in 2011 remained near or greater than the corresponding climatological values, with the exception of zonal advection. The zonal advection in 2011 indicates a major increase in dry tendencies (vertically

integrated values <-3 mm/d) extended from the 900mb to the upper troposphere that persisted from March to June (Figure 6b and 6c). Meanwhile, all other moisture transport sources in Figure 6 remained wetter than normal during the 2011 spring up until late summer, making the zonal advection of dry air solely responsible for the severe tropospheric drying during the drought onset, previously identified in Figure 3.

Over the NGP, zonal advection is the dominant moisture source year-round (vertically integrated values > 0.5 mm/d) with positive tendencies extending from above the PBL to 300mb (Figure 7a and 7c). The climatology of meridional advection reveals negative tendencies year-round throughout the troposphere, except during summer in lower tropospheric levels where the moisture convergence is noticeably larger overnight highlighting the northerly moisture transport

by the GP LLJ (Figures 7d and 7f). The vertical advection term is moderately positive during April and May and strongly negative during the rest of the year (Figures 7g and 7i). The transient terms reveal a vertical structure similar to that of the SGP with the strong negative tendencies confined to the May-September period (Figures 7j and 7l). Similar to the 2011 SGP drought, the 2012 NGP drought onset is marked by strong advection of dry air in April and May concentrated

in the lower free-troposphere, which lead to the large (<-1.5 mm/d) decline in vertically integrated moisture tendencies during that period (Figures 7b and 7c). Besides the horizontal advection, all other terms during the 2012 spring indicate normal or greater than normal moisture tendencies characterizing the zonal advection term as the large-scale source of tropospheric drying during the 2012 drought onset. In the 2012 summer, both the vertical and meridional

advection terms indicate large moisture divergence mostly due to considerable strengthening of the dry tendencies in the mid- and upper-troposphere from July onward.

To identify potential drivers of the spring-time tropospheric drying shown in sections 3.1 and 3.2, we decomposed the zonal, meridional, and vertical advection anomalies into their





thermodynamic, dynamic, and non-linear contributions (see section 2.2). The results for the
zonal advection term are presented in Figures 8 and 9 for the SGP 2011 and NGP 2012 events,
respectively. For both events, the contribution of dynamic and nonlinear terms to the anomalies
of zonal advection are considerably small (as compared to the thermodynamic term) during
spring and early summer and nearly zero throughout the year in both ERA-Interim and MERRA2
reanalysis. During spring and early summer, the thermodynamic term reveals large negative
moisture tendencies for both the SGP 2011 and NGP 2012 cases with the vertical structure of the
anomalous tendencies in the two reanalysis consistently agreeing with one another (Figures 8c,
8d, 9c, and 9d). Since the thermodynamic contribution is defined as the product of the
climatological zonal wind (featuring large westerlies at 700mb) and the gradient of anomalous
humidity, its variability is entirely controlled by the zonal gradient of $q_v$ anomalies. As a result,
the strong advection of dry lower- and mid-tropospheric tendencies during the 2011 and 2012
drought onsets were almost entirely forced by the zonal gradient of specific humidity, or more
simply, by a relatively drier troposphere in the US SW and Rockies located upwind of the SGP
and NGP.

### 3.3. **The relation between anomalous moisture advection and the spring and summer**
**dry/wet conditions**

The relationship between thermodynamic zonal moisture advection and anomalous dry/wet
conditions were investigated using single point lag/lead correlation maps between the tendency
term over the SGP and NGP and multiple atmospheric variables over the US (Figures 10 and
11). For both regions, the correlation between the MAM anomalies of the zonal thermodynamic
advection and specific humidity at 700 mb features a dipole pattern with strong positive
(negative) correlations over the US west and southwest (east and northeast) highlighting the
zonal gradient of humidity anomalies as the main driver of variability of the moisture term. At
the surface, the correlation maps for MAM precipitation and ET indicate a similar pattern with
significant positive correlations over the Rockies and US southwest and relatively weak negative
correlations over the eastern US for both regions (the magnitude of positive correlations are
stronger for SGP than NGP; Figures 10c, 10e, 11c and 11e). The positive correlations indicate
that the dry (wet) anomalies of ET and P over the upwind region are linked to the anomalous
moisture divergence (convergence) over the SGP and NGP.

The spring-time variability of thermodynamic advection over the GP is linked to the summer-
time surface and atmospheric conditions over the US interior plains. The correlation maps of JJA
$q_v$ for both SGP and NGP, indicate positive correlations over the central US, east of the Rockies,
and near zero correlations elsewhere over the US. The correlation between the MAM moisture
tendency in the SGP and JJA ET are strongly positive over the Rockies and central plains (Figure
10d). A similar correlation pattern exists for the NGP tendency and JJA ET with the band of
significant positive correlations extending from the eastern Rockies and central US to the US
Midwest and East (Figure 11d). Similar to ET maps, the correlations between the MAM moisture
term over both the south and north GP and the JJA precipitation anomalies are strongly positive
(> 0.45) over the US northern plains and Midwest and weakly positive over the southern plains



and northwestern US (Figures 10f and 11f). Similar correlation patterns were reproduced using
the CPC-gauged based precipitation as an independent observational data set in the lag/lead
correlations with the MAM moisture term anomalies in the SGP and NGP (see Figures S3 and
S4).

The strength and spatial patterns of the correlations between the moisture term and both MAM
and JJA precipitation (shown in Figures 10 and 11) signals a potentially significant relationship
between the MAM precipitation in the US SW and JJA precipitation in the GP. Using the CPC
precipitation, we calculated single-point correlations between the standardized anomalies of
MAM precipitation in the US SW and the JJA precipitation at each grid cell (Figure 12a). The
results indicate strong positive (> 0.3) correlations over the US west coast, Rockies, and northern
GP, weak positive correlations over the US mid-west, near zero correlations over Arizona and
SGP, and weak negative correlations over the US east and southeast. The contours of positive
correlations are especially strong over the NGP. The comparison of time series of JJA
precipitation anomalies over the NGP against the MAM precipitation anomalies in the US SW
(Figure 12b) indicates a strong coverability between the two time series during the 1979-2018
period with a correlation coefficient of 0.41 (significant at 1%). The correlation magnitude is
surprisingly large as compared to the near zero correlation between the standardized anomalies
of MAM and JJA precipitation in the NGP.

## 4. Discussion

Our analyses of the variability and vertical structure of MFC components during the SGP 2011
and NGP 2012 extreme droughts identified severe lower-free tropospheric drying over the US
SW, and the resultant dry zonal advection anomalies to the US GP in spring as the major drought
onset mechanism for both events. The influence of lower-tropospheric humidity on GP
precipitation grows continually in spring as the GP precipitation regime begins to shift from a
dominantly frontal precipitation regime in winter toward convective precipitation in summer.
Our results indicate that a drier lower free troposphere in the US GP, due to strong zonal
advection of dry air in spring, is associated with a sharp drop of RH above the PBL which
increases dry entrainment and decreases the buoyancy of a rising moist plume. The increased dry
entrainment would decrease precipitation during spring and early summer by limiting the
convective penetration depth and shifting the convection structure from predominantly deep
convective towers toward frequent shallow cumulus clouds (Derbyshire et al., 2004; Zhang et al.,
2010; Del Genio, 2012). For the SGP 2011 and NGP 2012 events, the suppressed convection in
spring and early summer was supported by the severe decrease (~30-40%) of specific cloud
liquid and iced water content above the PBL (Figures 3f and 4f) and the FCC in the upper
troposphere (Figures S1f and S2f). The strong control of the free-tropospheric humidity on
convective precipitation has already been demonstrated in both cloud-resolving model (CRM)
simulations as well as observational studies (Derbyshire et al., 2004; Sherwood et al., 2010;
Zhang et al., 2010; Zhuang et al., 2018). Meanwhile, the conventional convective
parameterization schemes tend to severely underestimate the sensitivity of moist convection to
environmental humidity largely due to underestimation of the turbulent entrainment of drier air





into the rising convective cells (Derbyshire et al., 2004; Del Genio, 2012). This underestimation would lead to overestimation of deep convection in climate models implementing convection schemes, and could be a potentially major source of uncertainty responsible for poor performance of the current dynamic models in predicting summer drought in the GP.

The temporal evolution of RH during the SPG 2011 and NGP 2012 droughts reveals a transition of the maximum dry anomalies of RH from the free-tropospheric levels in spring to the lower

troposphere and boundary layer in summer. A positive land-atmosphere feedback could facilitate this shift by perpetuating the initial dry land surface conditions in spring to the severe drying and warming in summer. In this mechanism, an anomalously lower precipitation and lower FCC would lead to a relatively drier surface and enhanced insolation in late spring. As a result, ET would decline steadily in the following months leading to a significant decrease in surface latent

heat flux (estimated about 50 w.m-2 for the 1988 summer by Lyon et al. 1995), which is largely balanced by an increase in upward sensible heat flux and air temperature. The hotter-drier surface would intensify the decline of boundary layer and lower tropospheric humidity causing further decrease of precipitation in summer. This feedback mechanism was found to be responsible for intensification of several extreme cases of summer drought and heat waves over

the US interior plains (Chang and Wallace, 1987; Hao, 1987; Namias, 1991; Lyon and Dole, 1995). The anomalous warming of the PBL in summer can also increase the difference between the surface temperature and dew point (T-$T_d$) resulting in elevation of the level of free convection (LFC), increase of convective inhibition energy (CIN), and suppression of deep convection (Hao, 1987; Myoung et al., 2010).

The breakdown of total MFC into its zonal, meridional, and vertical advection terms in our analysis shows the meridional and zonal advection terms to be the dominant sources of incoming moisture over the SGP and NGP, respectively. This is clear from the year-round strong positive tendencies of meridional advection over the SGP (confined to the lower troposphere; Figure 6d) and zonal advection in the free-tropospheric levels over the NGP (Figure 7a). While the role of

meridional advection of moisture from the Gulf of Mexico to the US interior plains has received extensive attention in the literature (Schubert et al., 1998; Weaver et al., 2008; Berg et al., 2015), the importance of zonal advection as a major moisture transport mechanism has been overlooked. In the case of the NGP 2012 drought, for example, the severe moisture divergence during the drought onset has been attributed to the dry anomalies of meridional moisture

advection as a result of weakening of the GP LLJ (Hoerling et al., 2013 and 2014). Our close examination of the moisture budget terms, however, rejects this suggestion by revealing higher than normal moisture convergence for the meridional term during both 2011 and 2012 events and attributing the observed tropospheric drying for the two events to the zonal advection term.

Further breakdown of moisture advection anomalies into their dynamic and thermodynamic

contributions suggests that the thermodynamic contribution was almost entirely responsible for the extreme dry anomalies of zonal advection during the SGP 2011 and NGP 2012 droughts. By definition, the thermodynamic contribution is driven by the gradient of $q_v$ and the dominance of zonal thermodynamic advection in the onset of the 2011 and 2012 events signifies the





importance of the west-east gradient of tropospheric moisture. The spatial patterns of MAM
climatology of $q_v$ indicate a relatively large meridional gradient where $q_v$ decreases sharply
moving northward from Mexico toward the GP and a smaller zonal gradient with higher $q_v$
values over the Rockies gradually decreasing in the eastward direction toward the NGP and US-
Midwest (Figure S5). Despite a relatively larger magnitude of the meridional gradient of
humidity, the zonal advection tendency becomes much larger (2 to 3 times over the SGP and 4 to
5 times over the NGP) than the meridional advection in the free tropospheric levels mainly due
to the large zonal (westerly) and the near zero meridional vectors of the horizontal wind over GP
at those levels. However, since the zonal gradient of moisture at the free tropospheric levels is
small, an anomalous dipole pattern (drier west-wetter east) or even a severe decline of $q_v$ over
the Rockies can change the direction of the climatological west-east moisture gradient diverting
the zonal thermodynamic advection tendency from its climatological values (strongly positive
over the NGP) to strong negative anomalies as large as those observed in the SGP 2011 and NGP
2012 MAM season.

The role of zonal thermodynamic advection in linking the dry/wet conditions over the GP and its
upwind region is further supported by the lag/lead correlation analysis between the moisture term
and multiple atmospheric and surface parameters in ERA-Interim reanalysis. Similar correlation
analysis applied to the CPC observed precipitation provided additional independent evidence
indicating that the MAM precipitation anomalies in the US SW region lead the variability of JJA
precipitation over the NGP (statistically significant at the 1% level).

## 5. Conclusions

We investigated the GP summer drought from a moisture budget perspective and looked at the
sub-daily, monthly, seasonal, and interannual variability of the moisture tendencies in two state-
of-the-art reanalysis. For the two extreme droughts (the SGP 2011 and NGP 2012) in our study
period, we found that a severe moisture divergence forced by the zonal advection tendency at the
lower free troposphere (850 mb to 600 mb) dominated the anomalously dry MFC at the early
stage of the droughts. The severe free-tropospheric drying resulted in a sharp drop of RH above
the boundary layer and an increase of dry entrainment which suppressed the deep convection
during spring, setting the stage for extremely dry summers. The anomalies of moisture
convergence were further decomposed into their dynamic and thermodynamic contributions with
the former isolating the impact of humidity gradient and the latter isolating the impact of wind
circulation. The results from ERA-Interim and MERRA2 consistently attributed the observed dry
anomalies of tropospheric moisture during the SGP 2011 and NGP 2012 drought onsets to the
thermodynamic contribution of the zonal advection tendency. The thermodynamic advection
tendency itself was strongly modulated by the spring-time conditions over the upstream region
(the US west and southwest) and significantly linked to the JJA precipitation and ET over the US
GP. The GP summer precipitation anomalies were found to be strongly correlated with MAM
precipitation anomalies in the US SW, suggesting the spring-time dry or wet anomalies over the
US SW and the Rockies to be a precursor of the drier or wetter summer over the US GP.





The results of this study provide a comprehensive picture of atmospheric moisture supply over the GP as well as the major drivers of strong moisture divergence during drought onset in the
GP. The identified relationship between spring conditions over the US SW and GP summer conditions can facilitate a better understanding of the hydrologic extremes over the GP and potentially improve the seasonal and sub-seasonal prediction skill.

## 6. **Authors Contribution**

AE designed the study and conducted the analysis in close collaboration with RF. Both authors
equally contributed in writing the manuscript, reviewing the results, and editing the paper.

## 7. **Acknowledgement**

The authors thank Robert Dickinson of University of California, Los Angeles for his constructive input and comments on this research. This study was supported by funding from the National Oceanic and Atmospheric climate Program office (NOAA-CPO), Modeling, Analysis,
Predictions, and Projections (MAPP) Program (NA17OAR4310123).

## 8. **Additional Information**

The authors declare no competing financial interests.






**Figures**

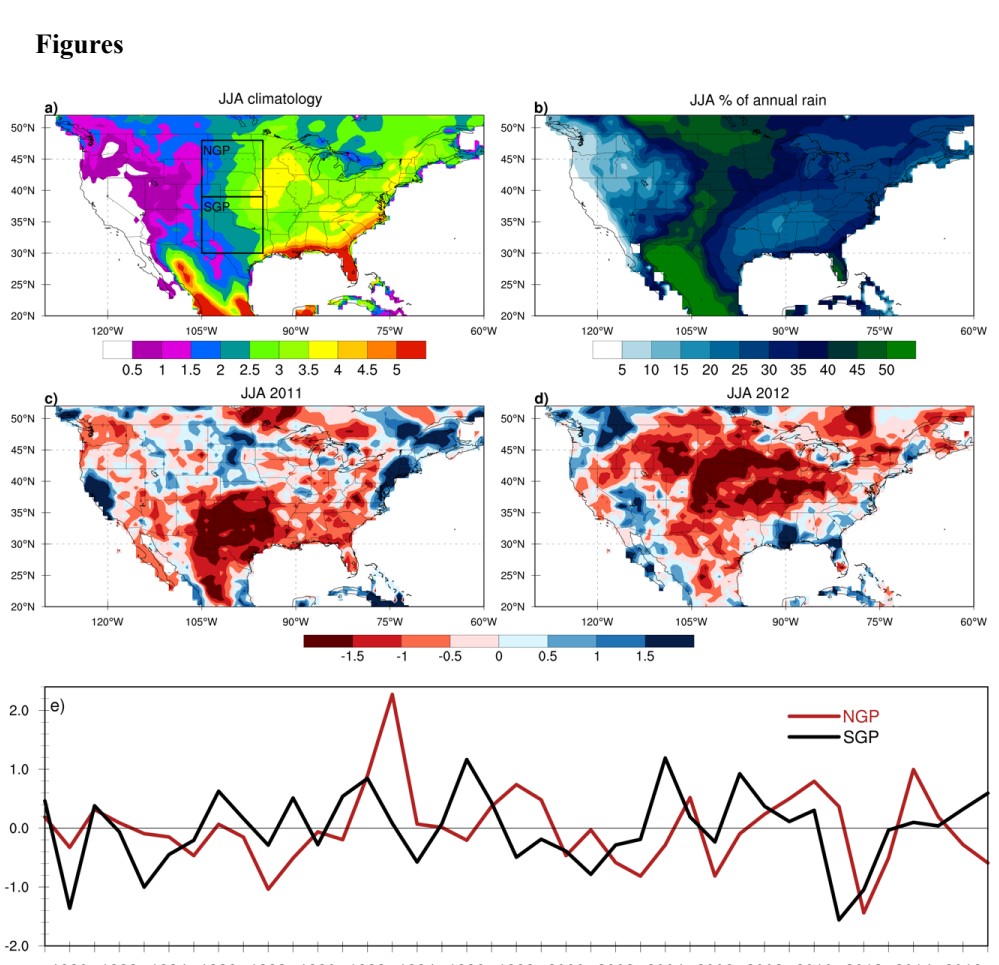

Figure 1. Spatial maps of JJA precipitation a) climatology (mm/d), b) percentage of the annual rain rate, and standardized anomalies (dimensionless) for the extreme droughts of c) 2011 and d) 2012. Monthly time series of the standardized anomalies of JJA precipitation are also shown (e) for the SGP and NGP regions (denoted by the boxes in a). The climatology and standardized anomalies were calculated using the CPC precipitation over the 1979-2018 period.






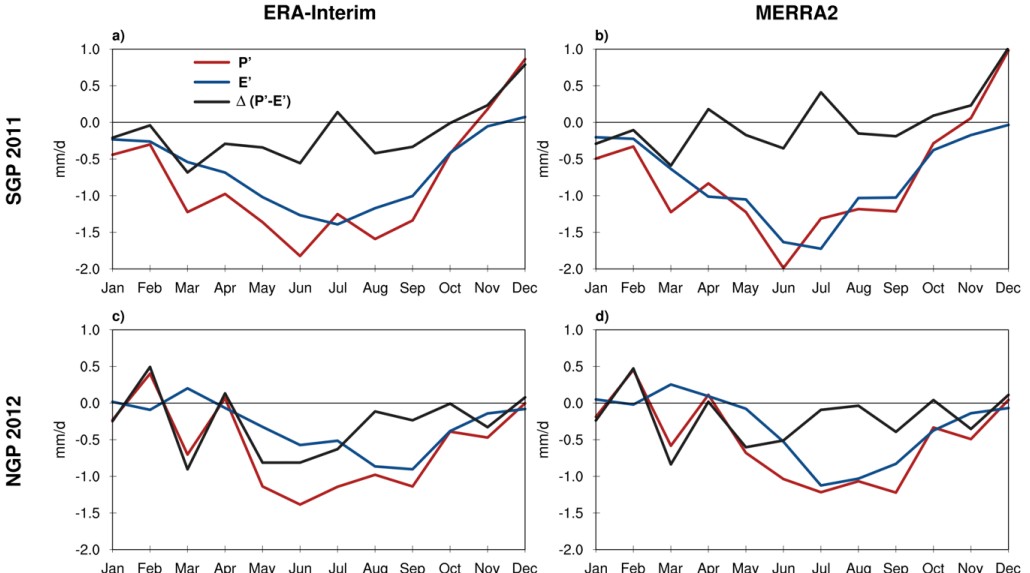

Figure 2. Annual cycle of precipitation (red), evapotranspiration (blue), and P-E (black)
       anomalies (mm/d) averaged over the SGP in 2011 (a and b) and the NGP in 2012 (c and d) using
       ERA-Interim (a and c: 1979-2018) and MERRA-2 (b and d: 1980-2018) reanalysis.





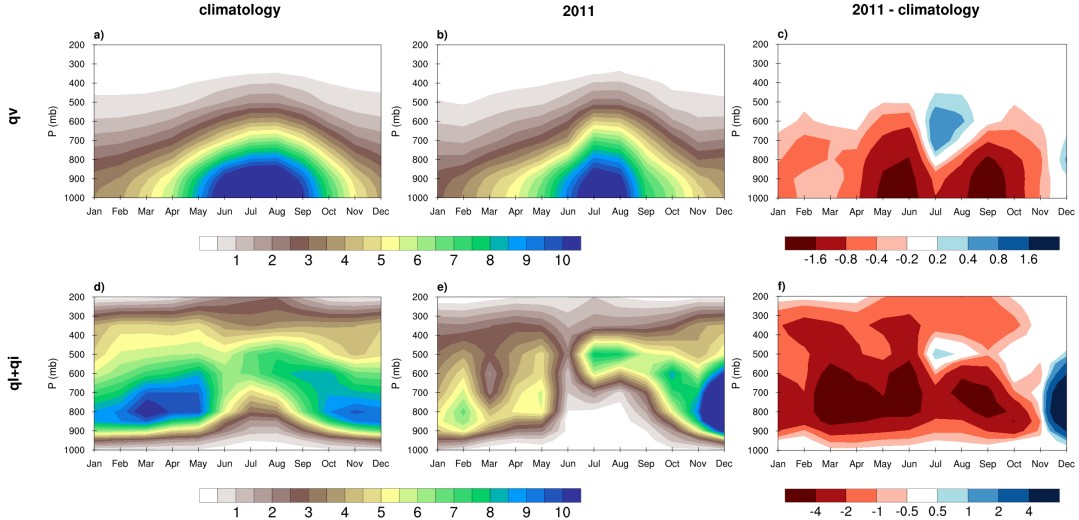


Figure 3. Hovmoller diagram of the vertical profile of the ERA-Interim specific humidity ($q_v$) (a, b, and c) and specific cloud liquid ($q_l$) and ice ($q_i$) water (d, e, and f) averaged over the US Southern Great Plains (30°-39° N and 95°-105° W) for the 1979-2018 climatology (a and d), 2011 (b and e), and the difference between the climatology and 2011 (c and f). The units for $q_v$,

$q_l$, and $q_i$ is g/kg.

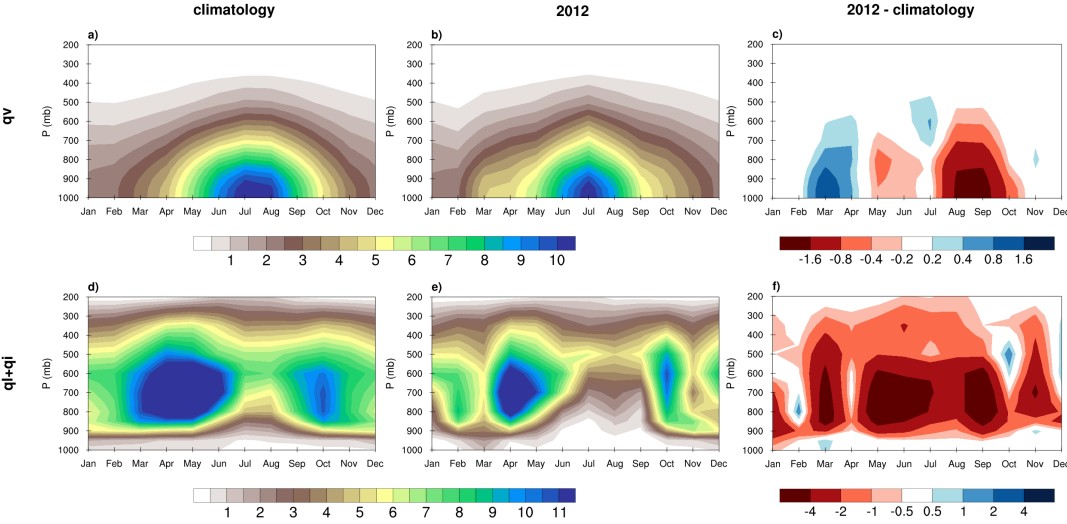

Figure 4. Same as Figure 3 but for the NGP (39°-48° N and 95°-105° W) in 2012.





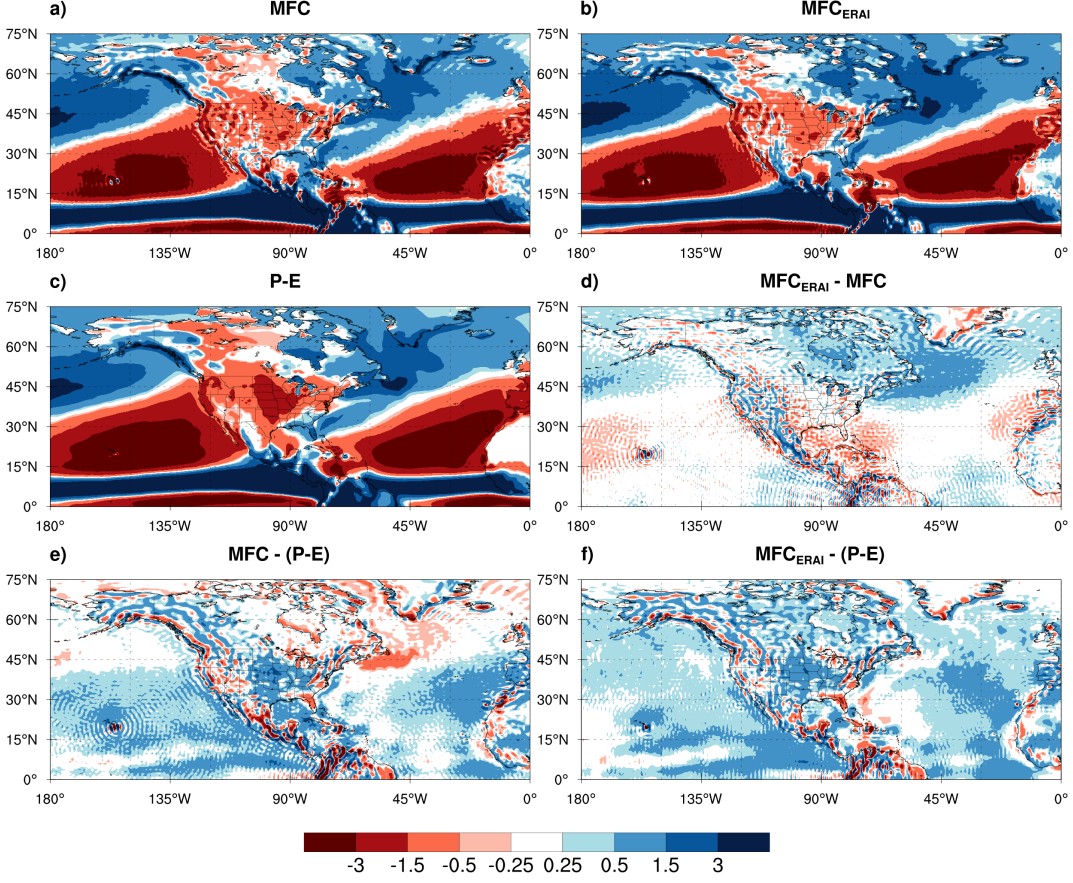

Figure 5. JJA climatology (1979-2018) of the vertically integrated MFC (mm/d) calculated diagnostically from the 6-hourly ERA-Interim output (a) and the monthly-mean MFC reported by ERA-Interim (b). JJA climatology of P-E (mm/d) has been also calculated from the ERA-Interim monthly outputs over the same period (c). The difference between the ERA-Interim reported and the calculated MFCs (d) represents the bias introduced by the numerical calculation of the budget terms in our analysis. The moisture budget imbalance is represented by subtracting the P-E climatology from those of the calculated MFC (e) and the ERA-Interim reported MFC (f).





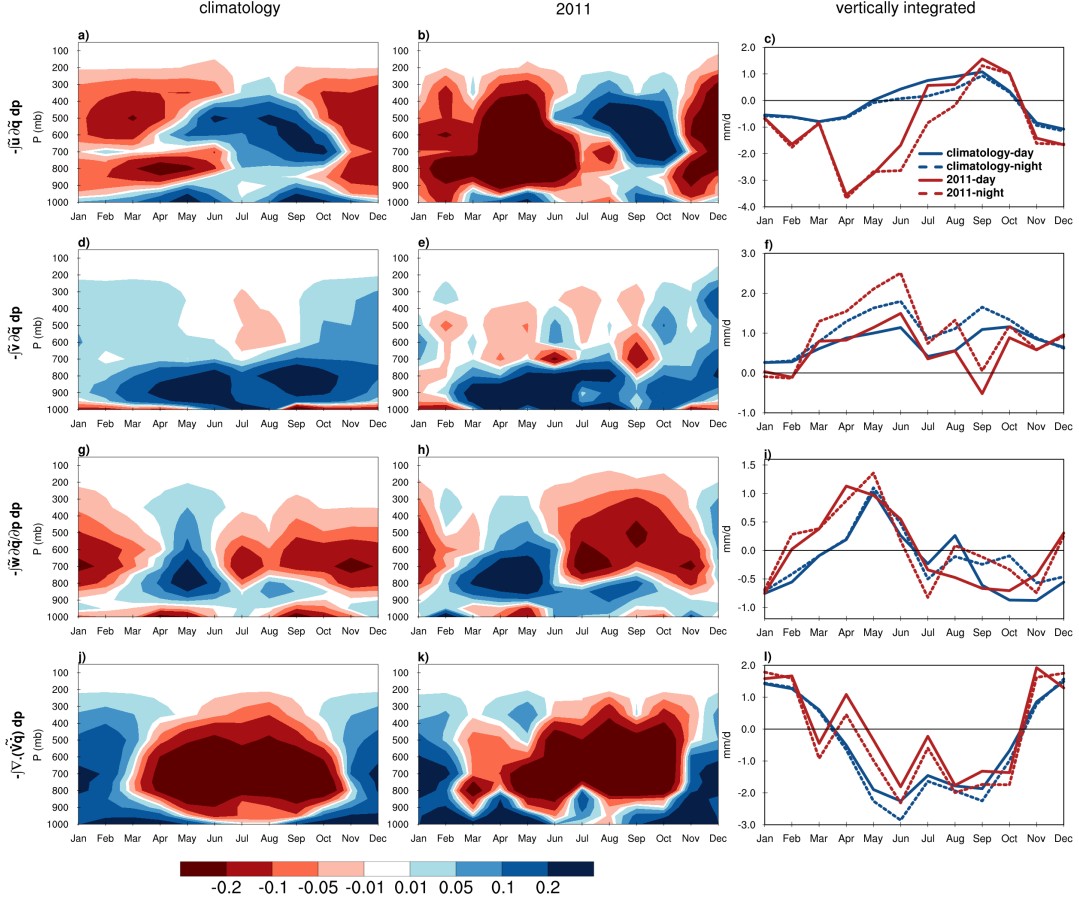

Figure 6. Hovmoller diagram of the vertical profile of the atmospheric moisture budget components averaged over the US Southern Great Plains for the 1979-2018 climatology (a, d, g, and j) and 2011 (b, e, h, and k). The 1st to 4th rows respectively represent the zonal advection, meridional advection, vertical advection, and horizontal transient terms in mm/day. The 3rd column (c, f, j, and l) represents the annual cycle of the corresponding terms (vertically integrated) for the climatology (blue) and 2011 (red) during the day-time (solid) and night-time (dashed) steps using 6-hourly ERA-Interim reanalysis.





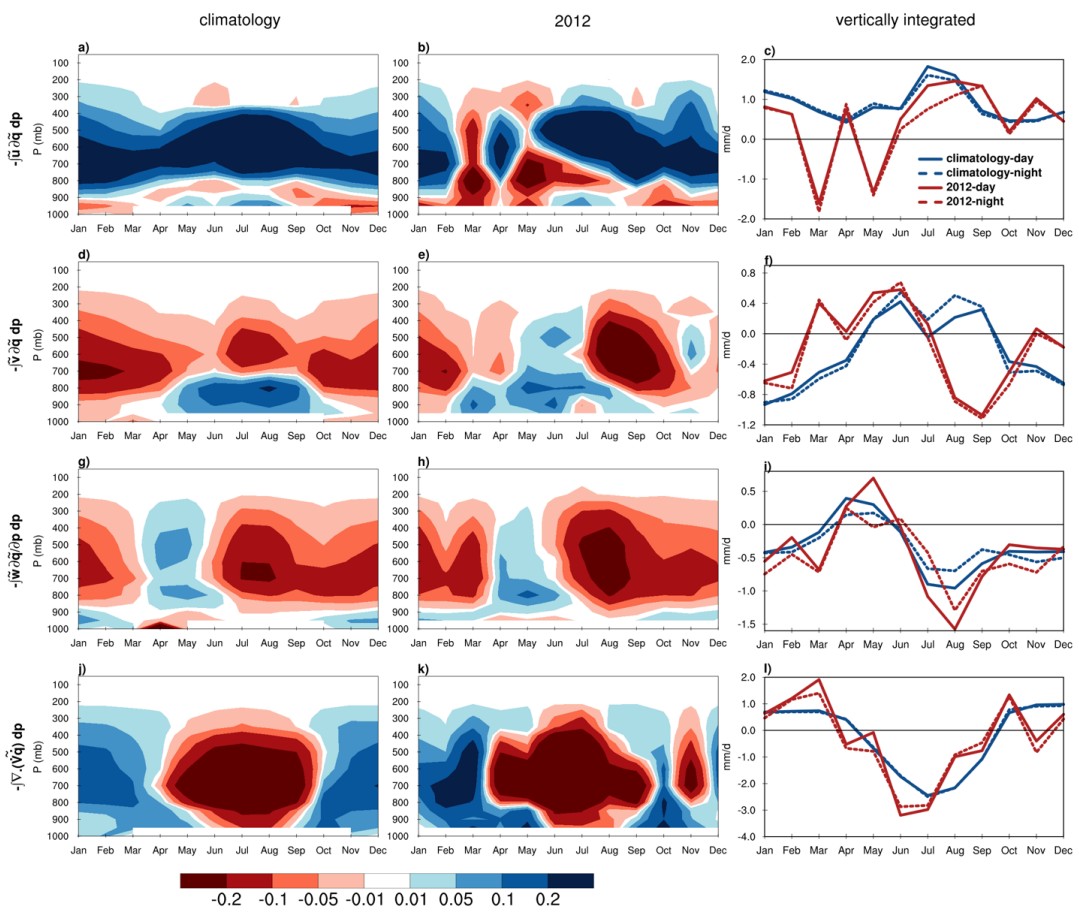


Figure 7. Same as Figure 6 but for the NGP in 2012.





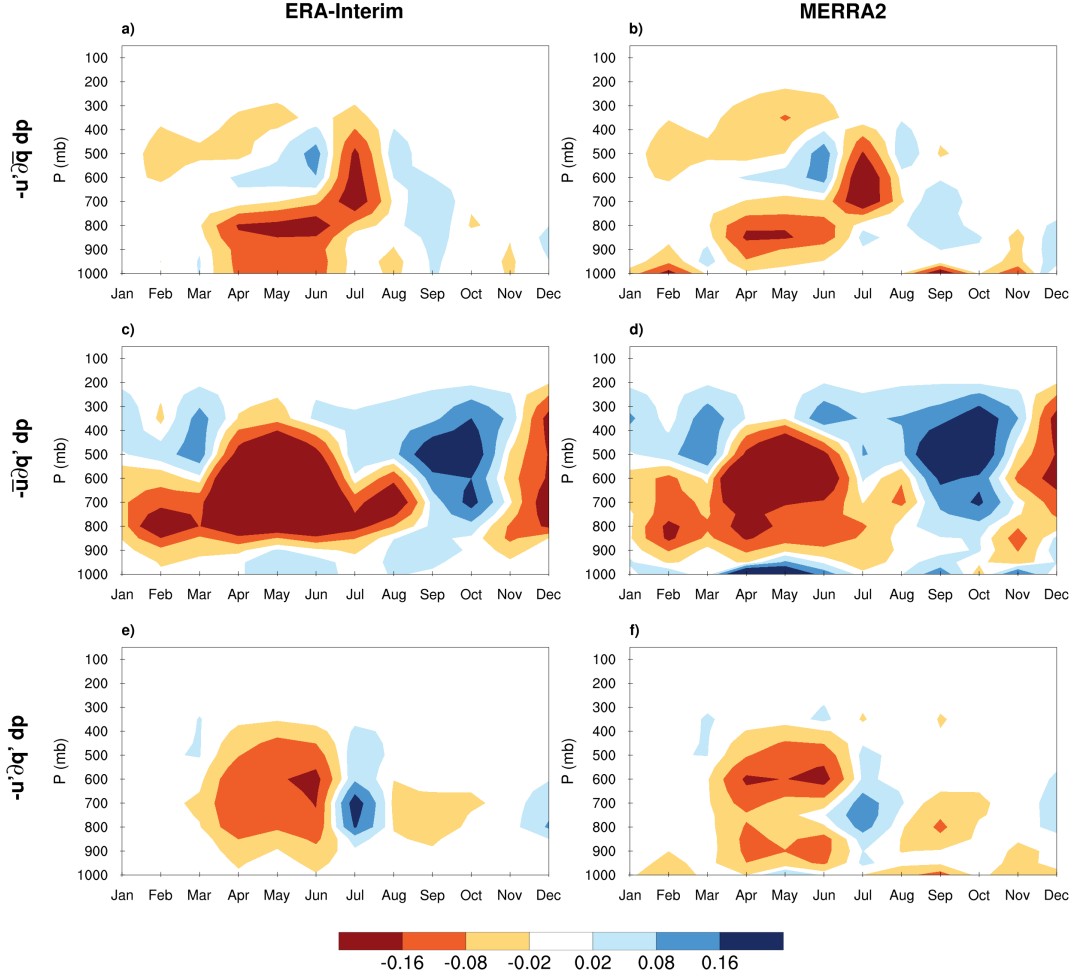

Figure 8. The Hovmoller diagram of the dynamic (a and b), thermodynamic (c and d), and non-
linear (e and f) contributions to the monthly anomalies of the zonal advection (mm/d) in ERA-
Interim (a, c, and e) and MERRA-2 (b, d, and f) over the SGP in 2011. The monthly anomalies
were calculated in respect to the 1979-2018 climatology for ERA-Interim and 1980-2018
climatology for MERRA2.





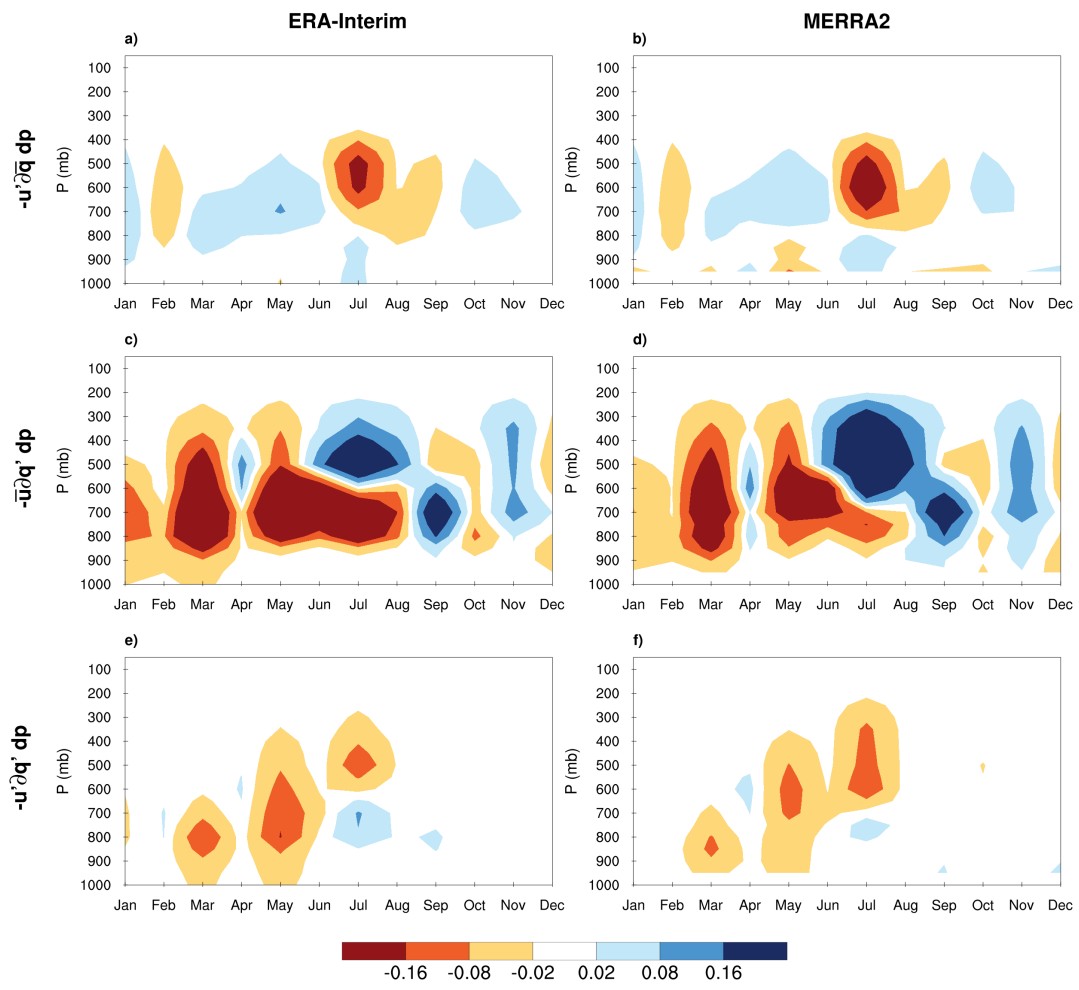


Figure 9. Same as Figure 8 but for the NGP in 2012.





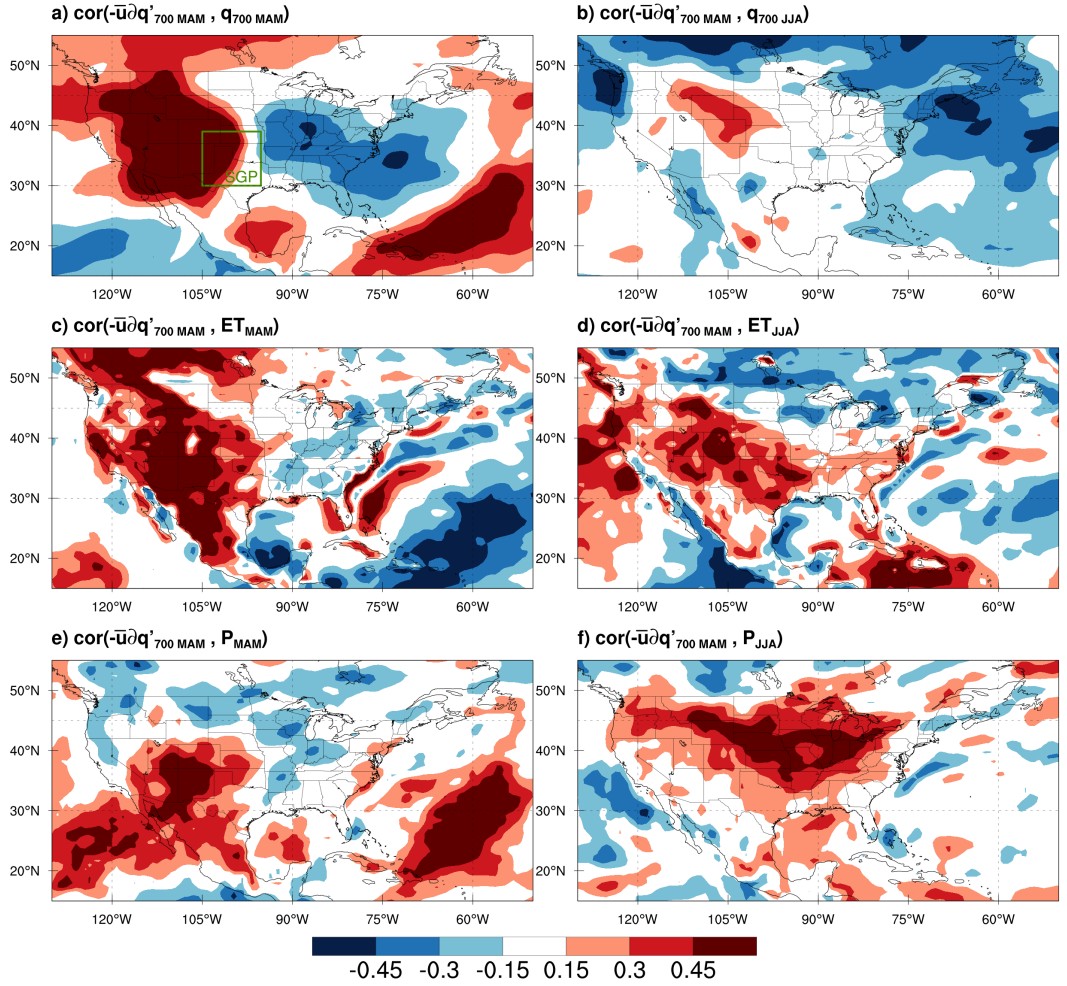

Figure 10. Single point correlation maps between the standardized time series (1979-2018) of the MAM zonal thermodynamic advection at 700 mb averaged over the SGP (the box in a) with the standardized anomalies of ERA- Interim specific humidity at 700 mb (a and b), evapotranspiration (c and d), and precipitation (e and f) for the MAM (a, c, and e) and JJA (b, d, and e) seasons. The correlation coefficients greater than 0.3 and 0.4 are statistically significant at the 10% and 2% levels, respectively (see section 2.6).





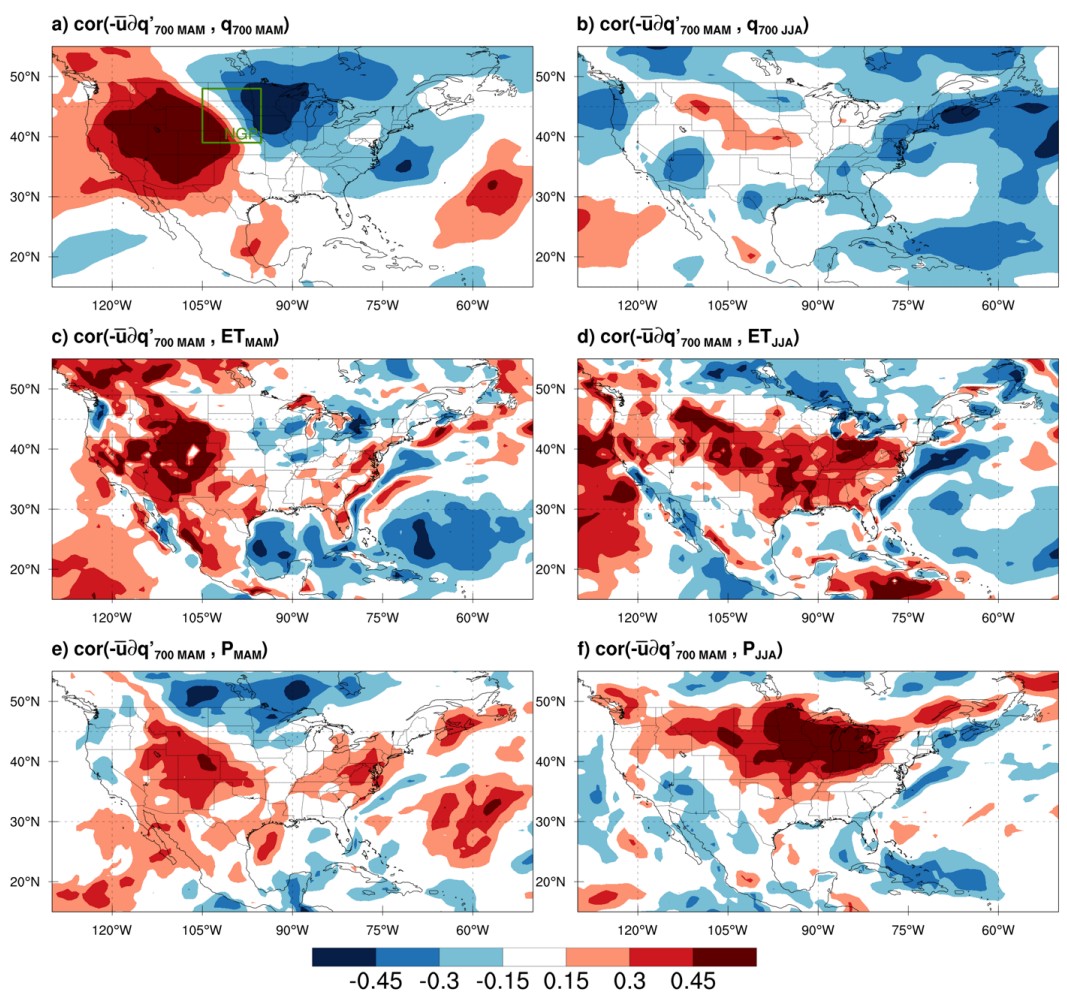

Figure 11. Same as Figure 10 but for the NGP.





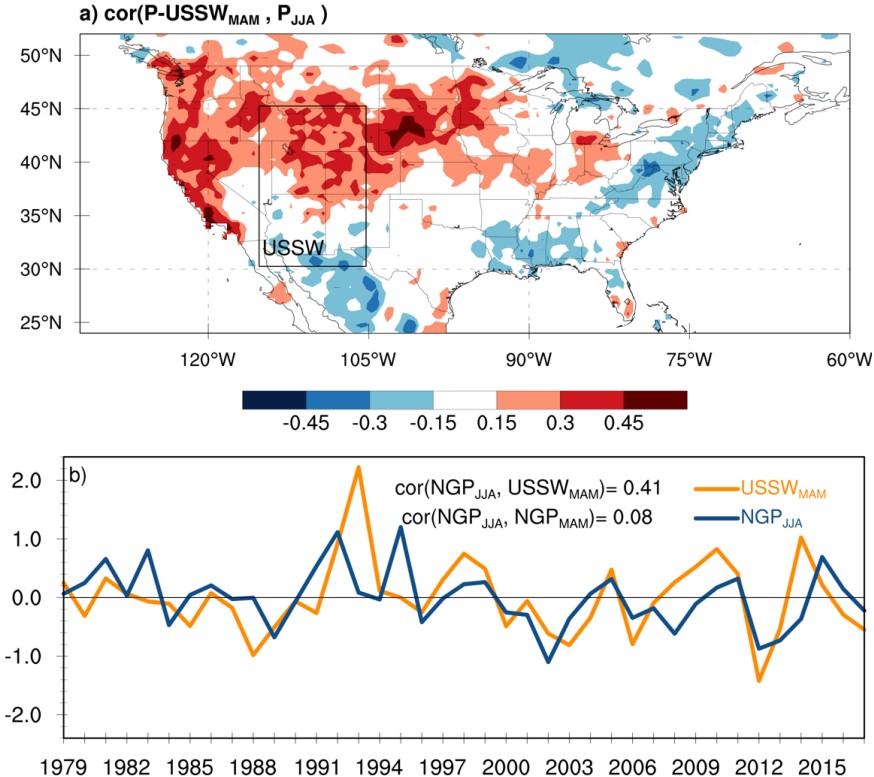

Figure 12. Single point correlation maps between standardized anomalies of MAM precipitation
over the US SW region (30°-45° N and 105°-115° W)  and JJA precipitation at each grid cell (a).
The time series of standardized anomalies of JJA precipitation over the NGP (blue) and MAM
precipitation in the US SW (yellow) are shown in panel b. The CPC gauged-based precipitation
from 1979 to 2018 was used to derive both the correlation map and time series.


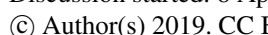



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
