# Peer review of "The role of dry zonal advection in summer drought onset over the US Great Plains"

_Atmospheric Chemistry and Physics, 2019_

## Referee Comment (RC1) · Anonymous Referee #1 · 28 May 2019

This study investigates the causes of the 2011 Southern Great Plains (SGP) and 2012 Northern Great Plains (NGP) droughts during JJA by performing a systematic atmospheric moisture budget analysis. The analysis reveals the key role of zonal advection of moisture in the preceding season (MAM) in leading to the drought condition in JJA. Through a simple correlation analysis (e.g. in Fig. 12), the study points out the importance of dry conditions in the SW US during MAM for droughts in the GP during JJA. The paper is overall neatly written and enjoyable to read, the results are cleanly presented. The finding on the importance of zonal thermodynamic moisture advection is interesting. The key finding on the importance of dry condition in the SW US during MAM for dry conditions in the GP in JJA, however, appears shaky, it needs to be

substantiated with further evidence. Please see my specific comments below.

1. Title: "Drier spring over the US Southwest as an important precursor of summer droughts over the US Great Plains"

The title appears to be based on Fig. 12, but the inference on the importance of dry spring over the U.S. Southwest as a precursor of summer droughts over the US Great Plains from Fig. 12 is not very convincing for the following reasons: 1) while the temporal correlation between JJA precipitation over the NGP and MAM precipitation over the SW US is statistically significant, it is based on all the cases, regardless of the sign and amplitude of the precipitation anomalies. If one focuses on dry cases only, the good correspondence between NGP precipitation during JJA and SW US precipitation during MAM is only shown for a limited number of cases (e.g. 1989, 2002, 2012, 2013), it is unclear whether the statistical relationship between the two regional precipitation still stands; 2) the SW US region is traditionally considered to cover the states of UT, CO, AZ and NM only. The SW US defined in Fig. 12 (black box in Fig 12a) appears to extend too far north. If limiting the SW US to cover the states of UT, CO, AZ and NM only, would the correlation results in Fig.12 change? 3) Fig.12 only suggests the relationship between MAM precipitation in the SW US and JJA precipitation in the NGP. It doesn't suggest any relationship for the JJA precipitation in SGP. It thus appears inappropriate to suggest that the MAM precipitation in the SW US can serve as a precursor for the precipitation in the GP as a whole.

2. Figure 2: the precipitation in the reanalyses are model dependent and are subject to deficiencies in the assimilation models used. How does the reanalysis precipitation in Fig. 2 compare with precipitation from observations (e.g. CPC gauge-based precipitation)?

3. Line 448: This study uses moisture budget analysis to show the importance of zonal moisture advection in MAM (due to dry anomaly in regions to the west) for both the 2011 and 2012 drought events. Droughts are known to be typically caused by

anomalous subsidence induced by upper-level anticyclonic circulation anomalies (e.g. Namias 1983). The 2011 and 2012 droughts also appear to have upper-level high anomalies occurring during their developing periods. Some discussions on how the zonal moisture advection may or may not connect to the upper-level high anomalies would be helpful.

Namias 1983: Some causes of United States drought. J. Climate Appl. Meteor.,22, 30–39.

4. Figures 10-12 are used to establish the connection between MAM zonal thermodynamic moisture advection and the development of GP droughts in the following JJA. Some discussions of possible physical processes by which the former (MAM zonal moisture advection) leads to the latter (JJA droughts in GP) would be helpful. The atmosphere does not have much memory: any atmospheric anomalies in MAM would presumably disappear in about 2 weeks. Is it possible that land plays some role (in sustaining the effect of MAM anomalies through JJA) here?

---

## Referee Comment (RC2) · Anonymous Referee #2 · 4 Jun 2019

This paper aimed to address the processes that lead to two summer droughts over US GPs in 2011 and 2012. The authors conducted a moisture budget analysis with two re-analysis products to show that zonal advection of anomalous moisture by mean winds is the dominant process that preceded and contributed to the two summer droughts.

While the moisture budget is suitable for the authors' aim, a major concern appears as to whether the resolution of the data used is high enough to close the budget. If the error term is comparable to the main terms (P-ET and moisture flux convergence), a further breakdown into different terms (advection, mass convergence, etc.) will be meaningless. This seems to be the case in the current manuscript. For example, as

indicated around Line 285, the imbalance in the budget is as large as 1.5mm/day over the US central plains, and is comparable to the maximum P-E deficit of 1-3mm/day (∼Line265) and the breakdown terms shown later. This large error is clear in Fig. 5 (e&f vs a&c) over the US GPs. To solve this issue, the authors should either show that at the current resolution the error terms are indeed much smaller compared to the breakdown terms presented in Fig. 6-7, or if that's not the case, try to use higher resolution data to reduce the error. In either case, it's necessary to include the error terms in Fig. 6-7.

Some minor issues: Section 2.1: the moisture budget equations are not clearly derived. The authors started by combining continuity and moisture equations to get the commonly used flux form of moisture equation (1), but then broke it down to the advection form in (2) to suit their aim, which seems circular and confusing. I urge the authors to rederive these equations (1-6), maybe by following some papers cited (such as Seager and Naomi 2013).

Line124: "the transient and stationary terms refer to the monthly mean and six-hourly departure " should be "the stationary and transient terms refer to the monthly mean and six-hourly departure"

Line 268/293/etc: the usage of "moisture flux convergence" is confusing, and doesn't seem to follow the convention. When P-E>0, the moisture flux divergence term in equilibrium should be negative and by convention is interpreted as "moisture flux convergence". Please clarify.

Line 388: 'coverability' –> covariability

---

## Author Comment (AC1) · 30 Jun 2019

We would like to thank the anonymous reviewer for taking the time to review our manuscript. We appreciate the constructive comments and suggested improvements and have revised the manuscript accordingly. Below are our point-by-point responses to the reviewer's comments.

Response to Anonymous Referee #1

"This study investigates the causes of the 2011 Southern Great Plains (SGP) and 2012 Northern Great Plains (NGP) droughts during JJA by performing a systematic atmo-

spheric moisture budget analysis. The analysis reveals the key role of zonal advection of moisture in the preceding season (MAM) in leading to the drought condition in JJA. Through a simple correlation analysis (e.g. in Fig. 12), the study points out the importance of dry conditions in the SW US during MAM for droughts in the GP during JJA. The paper is overall neatly written and enjoyable to read, the results are cleanly presented. The finding on the importance of zonal thermodynamic moisture advection is interesting."

Response: Thanks for the feedback.

"The key finding on the importance of dry condition in the SW US during MAM for dry conditions in the GP in JJA, however, appears shaky, it needs to be substantiated with further evidence. Please see my specific comments below.

1. Title: "Drier spring over the US Southwest as an important precursor of summer droughts over the US Great Plains". The title appears to be based on Fig. 12, but the inference on the importance of dry spring over the U.S. Southwest as a precursor of summer droughts over the US Great Plains from Fig. 12 is not very convincing for the following reasons:"

Response: We agree that the original title did not reflect the full content of the manuscript (which is primarily focused on the moisture budget analysis and the role of zonal thermodynamic advection in the onset of GP summer droughts) and we have revised the title as "The role of dry zonal advection in summer drought onset over the US Great Plains " to address this concern.

"1) while the temporal correlation between JJA precipitation over the NGP and MAM precipitation over the SW US is statistically significant, it is based on all the cases, regardless of the sign and amplitude of the precipitation anomalies. If one focuses on dry cases only, the good correspondence between NGP precipitation during JJA and SW US precipitation during MAM is only shown for a limited number of cases (e.g. 1989, 2002, 2012, 2013), it is unclear whether the statistical relationship between the

two regional precipitation still stands;"

Response: The correlation coefficient for dry-only samples (20 cases with PNGP-JJA <0.0) is considerably higher (0.54) than that calculated for all samples. It should be mentioned that the colors representing the NGP and US SW time series in Figure 12b legend were mistakenly reversed. We fixed the legend in the revised manuscript and apologize for any potential confusion caused by this mistake.

"2) the SW US region is traditionally considered to cover the states of UT, CO, AZ and NM only. The SW US defined in Fig. 12 (black box in Fig 12a) appears to extend too far north. If limiting the SW US to cover the states of UT, CO, AZ and NM only, would the correlation results in Fig.12 change?"

Response: We limited the southern and northern boundaries of the US SW region to the areas suggested by the reviewer and the results (Figure 1) indicate negligible changes as compared to those presented in Figure 12.

"3) Fig.12 only suggests the relationship between MAM precipitation in the SW US and JJA precipitation in the NGP. It doesn't suggest any relationship for the JJA precipitation in SGP. It thus appears inappropriate to suggest that the MAM precipitation in the SW US can serve as a precursor for the precipitation in the GP as a whole."

Response: We agree. The title and the text have been revised to address this comment.

"2. Figure 2: the precipitation in the reanalyses are model dependent and are subject to deficiencies in the assimilation models used. How does the reanalysis precipitation in Fig. 2 compare with precipitation from observations (e.g. CPC gauge-based precipitation)?"

Response: The MERRA2 precipitation used in Figure 2 is bias-corrected against observational precipitation and compares reasonably well against the CPC gauged-based precipitation over the US (Gelaro et al. 2017).

Gelaro, Ronald, Will McCarty, Max J. Suárez, Ricardo Todling, Andrea Molod, Lawrence Takacs, Cynthia A. Randles, et al. 2017. "The Modern-Era Retrospective Analysis for Research and Applications, Version 2 (MERRA-2)." Journal of Climate 30 (14): 5419–54. https://doi.org/10.1175/JCLI-D-16-0758.1.

"3. Line 448: This study uses moisture budget analysis to show the importance of zonal moisture advection in MAM (due to dry anomaly in regions to the west) for both the 2011 and 2012 drought events. Droughts are known to be typically caused by anomalous subsidence induced by upper-level anticyclonic circulation anomalies (e.g. Namias 1983). The 2011 and 2012 droughts also appear to have upper-level high anomalies occurring during their developing periods. Some discussions on how the zonal moisture advection may or may not connect to the upper-level high anomalies would be helpful."

Response: Point well-taken. We added such discussions in the revised manuscript (the last paragraph of the discussion section, L487): "... Previous studies have also identified an anomalous high and anticyclonic vorticity in the upper troposphere as an atmospheric driver of summer droughts over central North America (Chang and Wallace, 1987; Namias, 1991; Lyon and Dole, 1995; Cook et al., 2011; Donat et al., 2016; Fernando et al., 2016). For the two droughts of SGP 2011 and NGP 2012, the anomalies of 700 mb (and also 350 mb) height feature a dipole pattern with an anomalous low over the northwestern North America and an anomalous high over the southeastern US (Figure S5). This dipole pattern seems to be a part of a larger wave-like pattern extended over North Pacific and was also detected in correlation maps between the anomalies of (south and north) GP zonal thermodynamic advection and geopotential height at 700 mb (not shown). A comprehensive understanding of the large-scale drivers of the zonal moisture advection over the GP can provide valuable information about the underlying mechanisms and predictability of the GP summer droughts and is a focus of our ongoing research."

"4. Figures 10-12 are used to establish the connection between MAM zonal thermodynamic moisture advection and the development of GP droughts in the following JJA. Some discussions of possible physical processes by which the former (MAM zonal moisture advection) leads to the latter (JJA droughts in GP) would be helpful. The atmosphere does not have much memory: any atmospheric anomalies in MAM would presumably disappear in about 2 weeks. Is it possible that land plays some role (in sustaining the effect of MAM anomalies through JJA) here?"

Response: Yes. The free-tropospheric drying in spring and early summer acts as a drought onset mechanism and a positive land-atmosphere feedback would sustain/intensify the initial dry conditions toward the end of summer. We have included a full paragraph discussing this mechanism in detail (L426 to L442): "The temporal evolution of RH during the SPG 2011 and NGP 2012 droughts reveals a transition of the maximum dry anomalies of RH from the free-tropospheric levels in spring to the lower troposphere and boundary layer in summer. A positive land-atmosphere feedback could facilitate this shift by perpetuating the initial dry land surface conditions in spring to the severe drying and warming in summer. In this mechanism, an anomalously lower precipitation and lower FCC would lead to a relatively drier surface and enhanced insolation in late spring. As a result, ET would decline steadily in the following months leading to a significant decrease in surface latent heat flux (estimated about 50 w.m-2 for the 1988 summer by Lyon et al. 1995), which is largely balanced by an increase in upward sensible heat flux and air temperature. The hotter-drier surface would intensify the decline of boundary layer and lower tropospheric humidity causing further decrease of precipitation in summer. This feedback mechanism was found to be responsible for intensification of several extreme cases of summer drought and heat waves over the US interior plains (Chang and Wallace, 1987; Hao, 1987; Namias, 1991; Lyon and Dole, 1995; Saini et al., 2016). The anomalous warming of the PBL in summer can also increase the difference between the surface temperature and dew point (T-Td) resulting in elevation of the level of free convection (LFC), increase of convective inhibition energy (CIN), and suppression of deep convection (Hao, 1987; Myoung et al., 2010)."

[Figure]

[Figure]

[Figure]

**Fig. 1.** Figure 1. Same as Figure 12 but calculated for the US SW region (denoted with the box in a) with new boundaries (22D-42D N and 105D-114D W).

---

## Author Comment (AC2) · 30 Jun 2019

We thank the anonymous reviewer for taking the time to review our manuscript and providing useful comments. Below is our point-by-point response to the Reviewer's comments.

Response to Anonymous Referee #2

"This paper aimed to address the processes that lead to two summer droughts over US GPs in 2011 and 2012. The authors conducted a moisture budget analysis with two re-analysis products to show that zonal advection of anomalous moisture by mean winds

is the dominant process that preceded and contributed to the two summer droughts. While the moisture budget is suitable for the authors' aim, a major concern appears as to whether the resolution of the data used is high enough to close the budget. If the error term is comparable to the main terms (P-ET and moisture flux convergence), a further breakdown into different terms (advection, mass convergence, etc.) will be meaningless. This seems to be the case in the current manuscript. For example, as indicated around Line 285, the imbalance in the budget is as large as 1.5mm/day over the US central plains, and is comparable to the maximum P-E deficit of 1-3mm/day (âĹijLine265) and the breakdown terms shown later. This large error is clear in Fig. 5 (e&f vs a&c) over the US GPs. To solve this issue, the authors should either show that at the current resolution the error terms are indeed much smaller compared to the breakdown terms presented in Fig. 6-7, or if that's not the case, try to use higher resolution data to reduce the error. In either case, it's necessary to include the error terms in Fig. 6-7."

Response: 1) The impact of resolution on the accuracy of our numerical calculations was measured by the MFC(ERAInterim)-MFC(calculated) error metric, which indicates near zero errors over the GP (Figure 5d), significantly smaller than MFC or P-E. The budget imbalance, MFC-(P-E), is primarily due to the parameterization of moist processes and the moisture budget in the Reanalysis not being closed, and minimally affected by the resolution of the data used in our calculations (as shown by nearly identical imbalances over the GP for both MFC(ERAInterim) and MFC(calculated) in Figures 5e and 5f).

2) The magnitude of mean bias (the climatological imbalance in Figure 5) does not support the argument made in this comment as a large yet constant imbalance (zero variability) cannot account for even a small variability in the MFC. The key factor to look at here is the variance and whether or not the range of year-to-year variability of the residual is large enough to mask the anomalies of MFC or the breakdown terms. This was investigated by a more detailed analysis of the imbalance in Figure 1 (below).

For both regions, the climatological imbalance indicates mean bias magnitudes of about 1mm/d in April, May, and June and less than 0.5 mm/d for rest of the year and a standard deviation that remains in the 0 to 0.5 (mm/d) range year-round. For the two droughts of 2011 and 2012, the imbalance remains about equal to or less than 0.75 and 0.5 mm/d year-round, respectively, which is four to six times smaller than the zonal moisture advection anomalies during the onset of both droughts (2.5 to 3 mm/d) and cannot mask variability of the advection term.

"Some minor issues: Section 2.1: the moisture budget equations are not clearly derived. The authors started by combining continuity and moisture equations to get the commonly used flux form of moisture equation (1), but then broke it down to the advection form in (2) to suit their aim, which seems circular and confusing. I urge the authors to rederive these equations (1-6), maybe by following some papers cited (such as Seager and Naomi 2013)."

Response: Eq. 1 (the flux form of moisture budget) is replaced with the conservation of water vapor in the revised manuscript, as suggested by this comment.

"Line124: "the transient and stationary terms refer to the monthly mean and six-hourly departure " should be "the stationary and transient terms refer to the monthly mean and six-hourly departure""

Response: The order is fixed as suggested.

"Line 268/293/etc: the usage of "moisture flux convergence" is confusing, and doesn't seem to follow the convention. When P-E>0, the moisture flux divergence term in equilibrium should be negative and by convention is interpreted as "moisture flux convergence". Please clarify."

Response: Convergence is defined as -1*divergence (L277) and the moisture flux convergence (MFC) refers to the left side of Eq. 2 and represents the negative total moisture divergence flux which is consistent with the literature (e.g. Banacos and Schultz,

2005). To remove potential confusions, a detailed definition of MFC has been included in the revised manuscript (L126).

Banacos, Peter C., and David M. Schultz. "The use of moisture flux convergence in forecasting convective initiation: Historical and operational perspectives." Weather and Forecasting 20.3 (2005): 351-366, https://doi.org/10.1175/WAF858.1

"Line 388: 'coverability' –> covariability"

Response: Fixed as suggested.

Figure 1. The annual cycle of the moisture imbalance for the numerically calculated moisture tendencies (MFC(calculated)-(P-E)) averaged over the US a) SGP and b) NGP for the 1979-2018 climatology (solid black) overlaid with the one standard deviation envelope, the SGP in 2011 (solid blue), and the NGP in 2012 (dashed blue), versus the zonal moisture advection anomalies for the SGP in 2011 (solid red) and the NGP in 2012 (dashed red). All the units are in mm/d.

[Figure]

**Fig. 1.** Figure 1. The annual cycle of the moisture imbalance for the numerically calculated moisture tendencies (MFC(calculated)-(P-E))averaged over the US a) SGP and b) NGP for the 1979-2018 climatology(soli